# An evaluation of the tumor microenvironment through CALR, IL1R1, IFNB1, and IFNG to assess prognosis and immunotherapy response in bladder cancer patients

Lilong Liu[1]*[†], Zhenghao Liu[1†], Lei Fan[2†], Zhipeng Yao[1†], Junyi Hu[1], Yaxin Hou[1], Yang Li[1], Yuhong Ding[1], Yingchun Kuang[1], Ke Chen[1]*, Yi Hao[3]*, Zheng Liu[1]*

[1]Department of Urology, Tongji Hospital, Tongji Medical College, Huazhong University of Science and Technology, Wuhan, China; [2]Department of Urology, Xiangyang Central Hospital, Affiliated Hospital of Hubei University of Arts and Science, Xiangyang, China; [3]Scientific Research Center, Shanghai Public Health Clinical Center, Fudan University, Shanghai, China

*For correspondence:
ddluis1204@163.com (LL);
shenke@hust.edu.cn (KC);
haoyi@shaphc.org (YH);
lz2013tj@163.com (ZL)

[†]These authors contributed equally to this work

Competing interest: The authors declare that no competing interests exist.

## eLife Assessment

This study presents **useful** findings that explore the prognostic and immunotherapeutic relevance of specific immune-related genes (CALR, IL1R1, IFNB1, and IFNG) in the bladder cancer tumor microenvironment. While the analysis highlights potentially meaningful associations with survival and treatment response, the strength of evidence is **incomplete**, as some claims lack sufficient experimental or mechanistic validation. Further refinement and validation of the predictive models would enhance the impact and generalizability of the conclusions.

**Abstract** Immunogenic cell death (ICD) is a type of cell death sparking adaptive immune responses that can reshape the tumor microenvironment. Exploring key ICD-related genes in bladder cancer (BLCA) could enhance personalized treatment. The Cancer Genome Atlas (TCGA) BLCA patients were divided into two ICD subtypes: ICD-high and ICD-low. High ICD expression linked to increased immune cell infiltration and longer survival, but with potentially suppressed immune function. The high ICD group responded better to PD1-targeted therapy. A risk-scoring model with four ICD-related genes (CALR, IL1R1, IFNB1, IFNG) was validated across TCGA, GEO datasets, and tissue samples, showing higher risk score correlated with weaker anti-tumor immune function, more tumor-promoting elements, lower immunotherapy response rates, and shorter patient survival. This study connects ICD-related genes to BLCA prognosis and immune infiltration, offering a vital tool for personalized treatment guidance.

## Introduction

Bladder cancer (BLCA) is the 10th most common cancer globally, divided into non-muscle-invasive (NMIBC) and muscle-invasive (MIBC) types (*Sung et al., 2021*). NMIBC patients often have high recurrence after surgery, so postoperative treatment like bladder instillation with BCG (Bacillus Calmette-Guérin) or chemotherapy is recommended (*Sung et al., 2021*). MIBC is treated with radical

cystectomy, often combined with platinum-based neoadjuvant chemotherapy for better results (*Ma et al., 2023*). New immunotherapy using PD-1/PD-L1 inhibitors is showing promise for BLCA. In 2017, the US Food and Drug Administration approved atezolizumab and pembrolizumab for advanced cases intolerant to platinum-containing chemotherapy (*Suzman et al., 2019*). Response rates of 11.68% and 24.05% were seen with atezolizumab and pembrolizumab (*Necchi et al., 2017*; *Balar et al., 2017*). Chemotherapy can affect the tumor immune environment (*Galluzzi et al., 2017*), and ongoing trials are exploring combining immune checkpoint inhibitors (ICIs) with chemotherapy (ClinicalTrials.gov Identifier: NCT04383743, NCT04630730) (*Calleris et al., 2023*). Research in BLCA immunotherapy is in early stages, highlighting the need for innovative approaches combining chemotherapy and immunotherapy.

Immunogenic cell death (ICD) is a regulated cell death that, in an active immune system, triggers an adaptive immune response by exposing antigens from dying cells (*Galluzzi et al., 2020*). Certain chemotherapy drugs, like cisplatin (*Chen et al., 2023*) and gemcitabine (*McDonnell et al., 2015*), can induce ICD. While studied in preclinical BLCA models, the potential benefits of ICD-based therapies in BLCA lack conclusive evidence (*Xu et al., 2022*). It is crucial to explore the impact of ICD in clinical settings for BLCA treatment, be it through chemotherapy or immunotherapy. In this study, we discovered a connection between ICD-related genes and the prognosis and immune infiltration in BLCA patients using The Cancer Genome Atlas (TCGA)-BLCA, Gene Expression Omnibus (GEO) datasets, and tissue microarray staining. Our validated risk-scoring model effectively evaluates immune infiltration, prognosis, immunotherapy response, and drug sensitivity in BLCA, providing guidance for personalized treatment and future research.

## Results

### Stratification and pathway enrichment analysis of TCGA-BLCA based on ICD-related genes

We conducted unsupervised clustering analysis on 34 ICD-related genes to identify two ICD-associated subtypes. Subsequently, we defined ICD-high and ICD-low groups based on the expression levels of these genes and compared their ICD gene expression profiles (*Figure 1A*). A clinical heatmap demonstrated differences between the two subtypes in terms of Grade (*Figure 1B*). Furthermore, survival analysis (*Figure 1C*) indicated that the ICD-high group exhibited significantly longer survival times compared to the ICD-low group (p=0.039). Subsequently, we found that patients in the ICD-high group exhibited a significantly higher TMB (p<0.05) (*Figure 1—figure supplement 1A*). We also found that TP53, TTN, KMT2D, MUC16, and ARID1A are the most frequently mutated genes in both groups (*Figure 1—figure supplement 1B and C*).

Next, we examined the differential gene expression between the two subgroups, identifying 4321 DEGs (Differentially Expressed Gene), consisting of 2177 downregulated and 2144 upregulated genes. The heatmap displayed the top 50 upregulated and 50 downregulated genes (*Figure 1D and E*). To further identify pathways associated with immune activation specific to ICD-high and ICD-low groups, Gene Ontology (GO), Kyoto Encyclopedia of Genes and Genomes (KEGG), and Gene Set Enrichment Analysis (GSEA) were conducted. The results indicate a close association between elevated ICD expression and the activation of the immune microenvironment (*Figure 1F–L*).

### The immune characteristics within the TME in ICD-high and ICD-low groups

We employed the ESTIMATE algorithm for analysis. The ICD-high group exhibited higher ESTIMATE scores, immune scores, and stromal scores compared to the ICD-low group, while the ICD-high group demonstrated lower tumor cell purity (*Figure 2A–D*). Further analysis of immune cell infiltration between the two subtypes revealed significantly higher levels of T cells CD8+, T cells CD4+ memory activated, and T cells follicular helper in the ICD-high group, while Tregs displayed the opposite trend (*Figure 2E*). Furthermore, we compared the expression of immune function-related pathways, immune checkpoint genes, and HLA-related genes between the two subtypes. The results indicate that the ICD-high group exhibited elevated expression in nearly all 13 immune pathways (*Figure 2F*). In the eight immune checkpoint genes, PDCD1LG2, CTLA4, PDCD1, CD274, HAVCR2, LAG3, and TIGIT had higher expression levels, while SIGLEC15 had lower expression

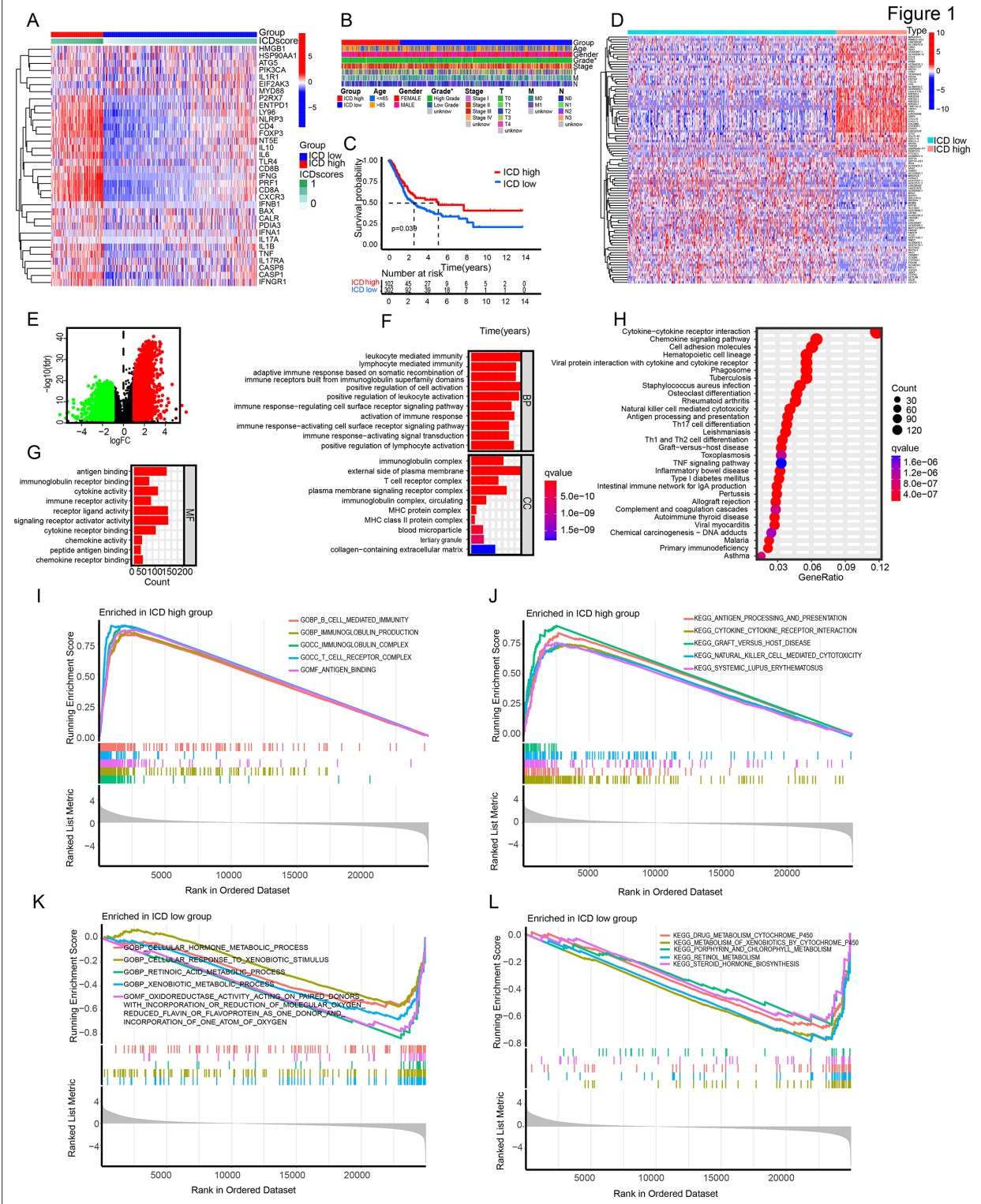

**Figure 1.** Stratification and pathway enrichment analysis of TCGA-BLCA based on ICD-related genes. (**A–C**) Gene expression heatmap (**A**), clinical heatmap (**B**), and survival analysis (**C**) of ICD-high and ICD-low groups. (**D, E**) Differential gene heatmap (**D**) and volcano plot (**E**) of ICD-high and ICD-low groups. (**F, G**). GO enrichment analysis. (**H**) KEGG enrichment analysis. (**I–L**) GSEA enrichment analysis.

The online version of this article includes the following figure supplement(s) for figure 1:

**Figure supplement 1.** Somatic mutations in ICD-high and ICD-low groups.

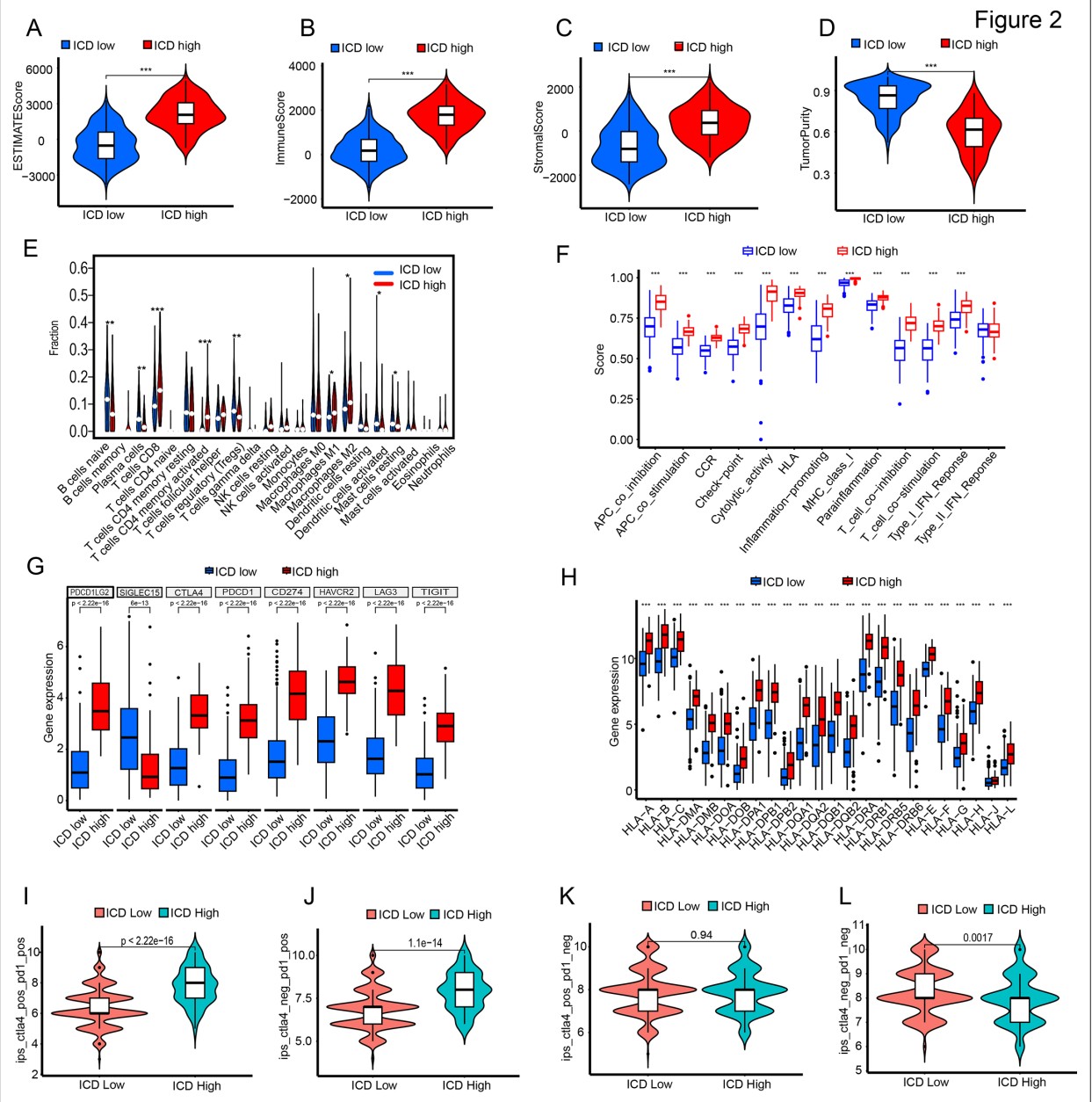

**Figure 2.** The immune characteristics within the tumor microenvironment vary between ICD-high and ICD-low groups. ESTIMATE scores (**A**), immune scores (**B**), stromal scores (**C**), and tumor cell purity (**D**) of ICD-high and ICD-low groups. (**E**) Immune cell infiltration in the two subgroups. (**F–H**) Expression profiles of the two subgroups in 13 immune-related pathways (**F**), 8 immune checkpoints (**G**), and HLA gene family (**H**). I. Efficacy of immunotherapy in the two subgroups.

(*Figure 2G*). Among the 24 HLA-related genes, the ICD-high group had higher expression levels for almost all genes (*Figure 2H*). These results suggest that ICD-high patients have a significantly higher quantity of tumor-infiltrating immune cells, but they appear to be in a state of functional exhaustion. Thus, we evaluated the effects of ICIs PD1 and CTLA4 on the ICD-high and ICD-low groups. Treated with PD1 and CTLA4 or PD1 alone, the ICD-high group had better outcomes. With CTLA4 alone, both subtypes had similar results. Without PD1 or CTLA4 treatment, the ICD-high group had significantly worse outcomes than the ICD-low group (*Figure 2I–L*). Demonstrating that targeting PD-1 is crucial to enhance immune cell functionality and improve the prognosis of ICD-high patients.

## Construction and validation of risk-scoring model

In Cox univariate analysis, we identified 4 ICD-related genes significantly associated with OS of patients: CALR (p=0.003), IFNB1 (p=0.037), IFNG (p=0.022), and IF1R1 (p=0.047) (*Figure 3A*). Subsequently, these four ICD-related genes were tested and selected in the LASSO regression analysis for constructing the predictive model (*Figure 3B and C*). Subsequently, patients are categorized into high-risk and low-risk groups according to the median (the boundary value is 4.372) of the risk scores. Survival analysis indicated that, in both the TCGA-BLCA dataset and the GSE13507 dataset, the high-risk group exhibits significantly shorter survival times than the low-risk group (p<0.05) (*Figure 3D and H*). Next, we analyzed somatic mutations and generated waterfall plots for the top 20 mutated genes (*Figure 3—figure supplement 1A and B*) and calculated the TMB for both groups. Among the ICD-high-risk group and ICD-low-risk group, TP53, TTN, KMT2D, MUC16, and ARID1A were the most frequently mutated genes, but the TMB does not differ between the ICD high-risk group and the ICD low-risk group (*Figure 3—figure supplement 1E*). Survival analysis also indicated longer survival times in the high TMB group (p<0.001), with the best prognosis observed in the low-risk + high TMB group and the worst prognosis in the high-risk + low TMB group (*Figure 3—figure supplement 1C and D*).

We further conducted a comprehensive analysis using the TCGA-BLCA dataset (*Figure 3E–G*) as the training set and the GSE13507 dataset (*Figure 3I–K*) as the validation set. Based on the median of the risk scores, we categorized the datasets from TCGA and GEO into high-risk and low-risk groups. We analyzed the correlation between risk scores and survival status, showing that higher risk scores increased the risk of patient mortality. Additionally, we examined the expression of the four genes in the risk signature, finding higher expression of CALR and IL1R1 in the high-risk group, and higher expression of IFNB1 and IFNG in the low-risk group (*Figure 3G and K*). Overall, these results suggest a less favorable prognosis for high-risk patients.

## The association between risk scores and clinical features

We analyzed the association between the four genes composing the risk score and patient prognosis (*Figure 4A–D*). Patients with low expression of CALR, low expression of IF1R1, high expression of IFNB1, and high expression of IFNG had better prognosis, which is consistent with CALR and IF1R1 being lowly expressed and IFNB1 and IFNG being highly expressed in the low-risk group (*Figure 3G and K*). The clinical heatmap demonstrates differences between the two subtypes in terms of stage and grade (*Figure 4E*). In the high-risk group, there is a higher proportion of stage III and stage IV cases, while stage I and stage II cases are less common (*Figure 4F*). Additionally, the high-risk group has a higher proportion of high-grade cases and a lower proportion of low-grade cases (*Figure 4G*).

To aid clinicians in making more intuitive predictions of individual patient survival probabilities at 1, 3, and 5 years, we created a nomogram that incorporated gender, grade, age, and risk score (*Figure 4H*). The AUC (Area Under the Curve) values for the 1-year, 3-year, and 5-year ROC(Receiver Operating Characteristic Curve) curves are 0.630, 0.635, and 0.651, respectively (*Figure 4I*). The AUC values for risk score, age, gender, and stage are 0.630, 0.669, 0.488, and 0.637, respectively (*Figure 4J*). Additionally, we found that the risk score can serve as an independent prognostic factor (p<0.001) (*Figure 4K and L*). This suggests that the risk score is an effective indicator for predicting patient prognosis.

We checked risk scores and survival in different subgroups. Higher risk scores were seen in advanced stages (high-grade, stage III, stage IV, and M1 cases), suggesting an association with advanced disease (*Figure 4—figure supplement 1A–G*). Analyzing the relationship with patient prognosis, significant survival differences were noted between high-risk and low-risk groups in most subgroups (*Figure 4—figure supplement 1H–S*). This implies that high-risk classification is linked to shorter survival, especially in advanced stages compared to early stages. To explore the link between risk scores and the immune microenvironment, we used the ESTIMATE algorithm. The high-risk group showed significantly lower immune scores and ESTIMATE scores than the low-risk group, while there was no notable difference in stromal scores between the two groups (*Figure 4M*). In the context of 13 immune-related pathways, the high-risk group demonstrated lower pathway activity (*Figure 4—figure supplement 2A*). Further analysis of immune cell infiltration between the two subtypes revealed that the high-risk group had significantly lower levels of T cells CD8, T cells CD4 memory activated, M1, and NK cells resting (*Figure 4N*). Correlation analysis between risk scores and immune cells indicated a negative

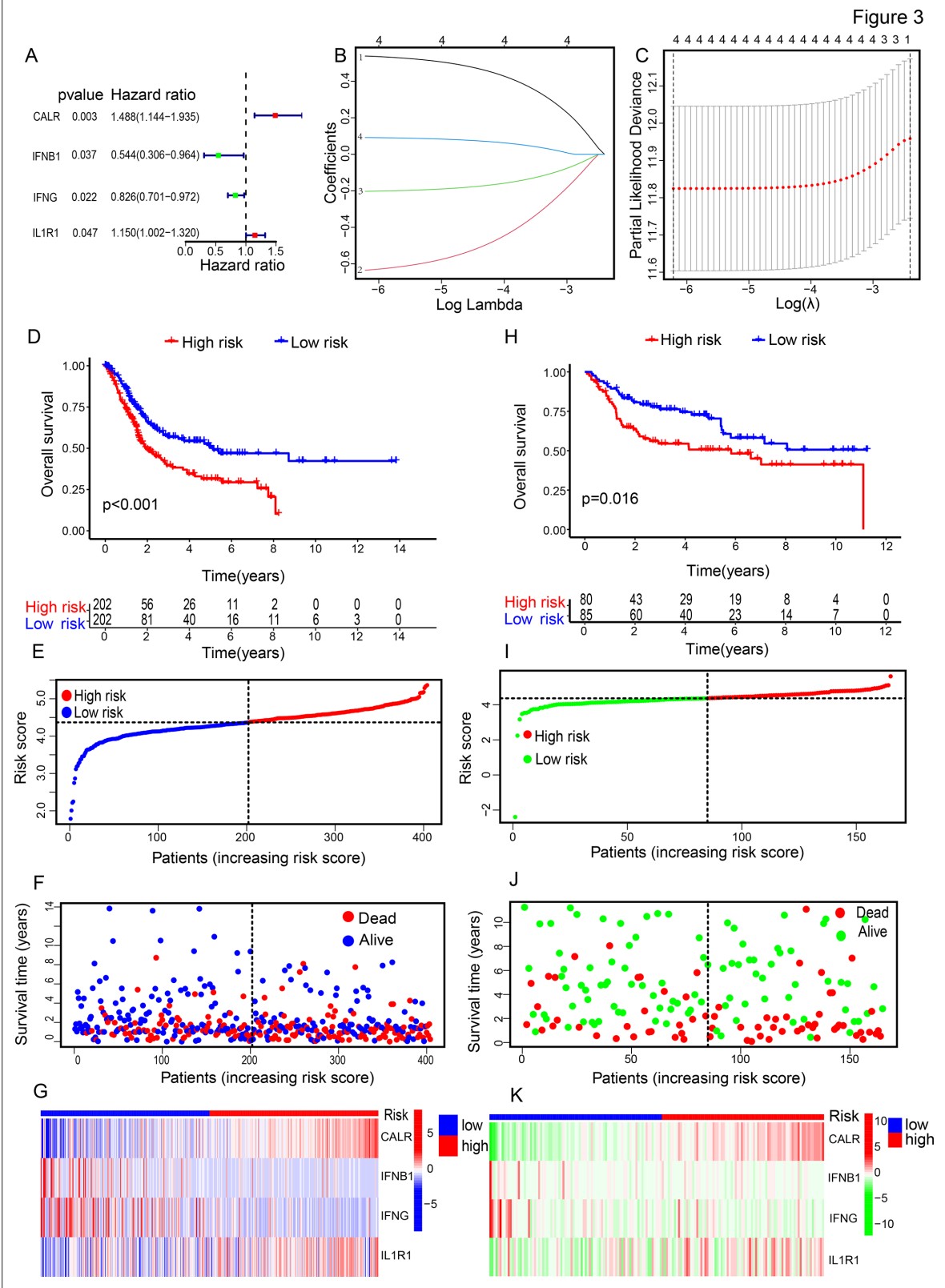

**Figure 3.** Construction and validation of risk-scoring model. (**A**) Hazard ratio of four ICD-related genes identified by Cox regression analysis. (**B, C**) LASSO regression analysis for constructing the prognostic model. (**D, E**) Survival analysis of high-risk and low-risk groups in TCGA (**D**) and GEO (**E**) databases. Analyzed the correlation between risk scores and patients, along with four key genes, in both TCGA (**F–H**) and GEO databases (**I–K**).

*Figure 3 continued on next page*

*Figure 3 continued*

The online version of this article includes the following figure supplement(s) for figure 3:

**Figure supplement 1.** Prognostic analysis of somatic mutations in high-risk and low-risk groups.

correlation with anti-tumor immune cells. Conversely, it showed a positive correlation with immune inhibitory cells (*Figure 4—figure supplement 3A–P*, *Figure 4—figure supplement 2B*). In addition, a significant correlation was observed between the 4 individual genes (CALR, IFN1B, IFNG, and IL1R1) comprising the risk score and immune cells, immune checkpoints, or immune regulatory pathways (*Figure 4—figure supplement 4A–P*, *Figure 4—figure supplement 5A–C*). These analyses suggest that higher risk scores are linked to weaker anti-tumor immune function and more tumor-promoting elements, partly explaining the shorter survival in the high-risk group.

## The association between the risk scores and the sensitivity to chemotherapy and immunotherapy

One of the primary factors inducing ICD is chemotherapy drugs, and targeted therapy plays a crucial role in BLCA treatment. Therefore, we further explored the correlation between the risk scores and the sensitivity to chemotherapy drugs and targeted therapies from TCGA. The high-risk group exhibited higher sensitivity (lower IC50 values) to sorafenib, epothilone B, docetaxel, elesclomol, MG132, lapatinib, and others. Conversely, the ICD-high risk group showed increased resistance (higher IC50 values) to lenalidomide, sunitinib, gefitinib, methotrexate, and camptothecin (*Figure 5A–P*). Furthermore, we investigated the prognostic characteristics of both groups in response to immunotherapy. The outcomes indicate that without anti-CTLA4 and anti-PD1 treatment, there was no significant difference in survival time between the groups. However, with at least one form of immunotherapy, the low-risk group had a significantly higher response rate than the high-risk group (*Figure 5Q–T*). These findings highlight the clinical potential of the risk score in predicting sensitivity to various therapies and guiding treatment decisions for better outcomes.

## Validate the risk-scoring model through BLCA tissue arrays and analyze the gene expression with single-cell sequencing

To validate the reliability of the aforementioned results, we conducted an analysis of the expression levels of CALR, IFN1B, IFNG, and IL1R1 and their associations with prognosis and immune function using BLCA tissue arrays. Immunofluorescence staining revealed that CALR expression in tumor tissues was higher than in normal tissues, while the other three genes, IL1R1, IFNB1, and IFNG, showed no significant differences (*Figure 6A and B*, *Figure 6—figure supplement 1A*). This observation is consistent with the results from the TCGA-BLCA dataset exported using the GEPIA 2 tool (*Figure 6—figure supplement 1B–E*). Patients were divided into high- and low-expression groups based on the median expression levels (immunoreactive score (IRS)) of CALR, IFN1B, IFNG, and IL1R1. It was found that patients in the CALR high-expression group and IL1R1 high-expression group had shorter survival times, while those in the IFNB1 high-expression group and IFNB high-expression group had longer survival times (*Figure 6C*), consistent with the results of the TCGA-BLCA dataset (*Figure 4A–D*). Patients' risk scores were similarly calculated using the risk-scoring model: Risk score=CALR-IRS*(0.5378)+IFN1B-IRS *(–0.6349)+IFNG-IRS *(–0.2028)+IL1R1-IRS *(0.0919). The high-risk group (50% vs 50%) had significantly shorter survival times than the low-risk group (*Figure 6D*), corroborating the previous results (*Figure 3D and E*). More importantly, we found a significant positive correlation between the risk score and CD39, a molecule we previously confirmed to be associated with immune suppression (*Liu et al., 2022*). Additionally, the risk score showed a significant positive correlation with immune exhaustion (CD8+LAG3+), while having a negative correlation with CD8+ T cell infiltration (the IRS are based on our previous research *Liu et al., 2022*). However, there was no correlation with tumor volume and T-stage (*Figure 6E–I*). This suggests a strong link between higher risk scores and immune suppression.

Finally, using our previous single-cell sequencing data (*Liu et al., 2022*) and the public dataset GSE135337, we analyzed the cellular expression patterns of CALR, IFN1B, IFNG, and IL1R1. CALR is expressed in nearly all cells in the TME, IL1R1 is mainly expressed in tumor-associated fibroblasts and endothelial cells, and the expression levels of IFNG and IFNB1 are minimal, primarily by T lymphocytes.

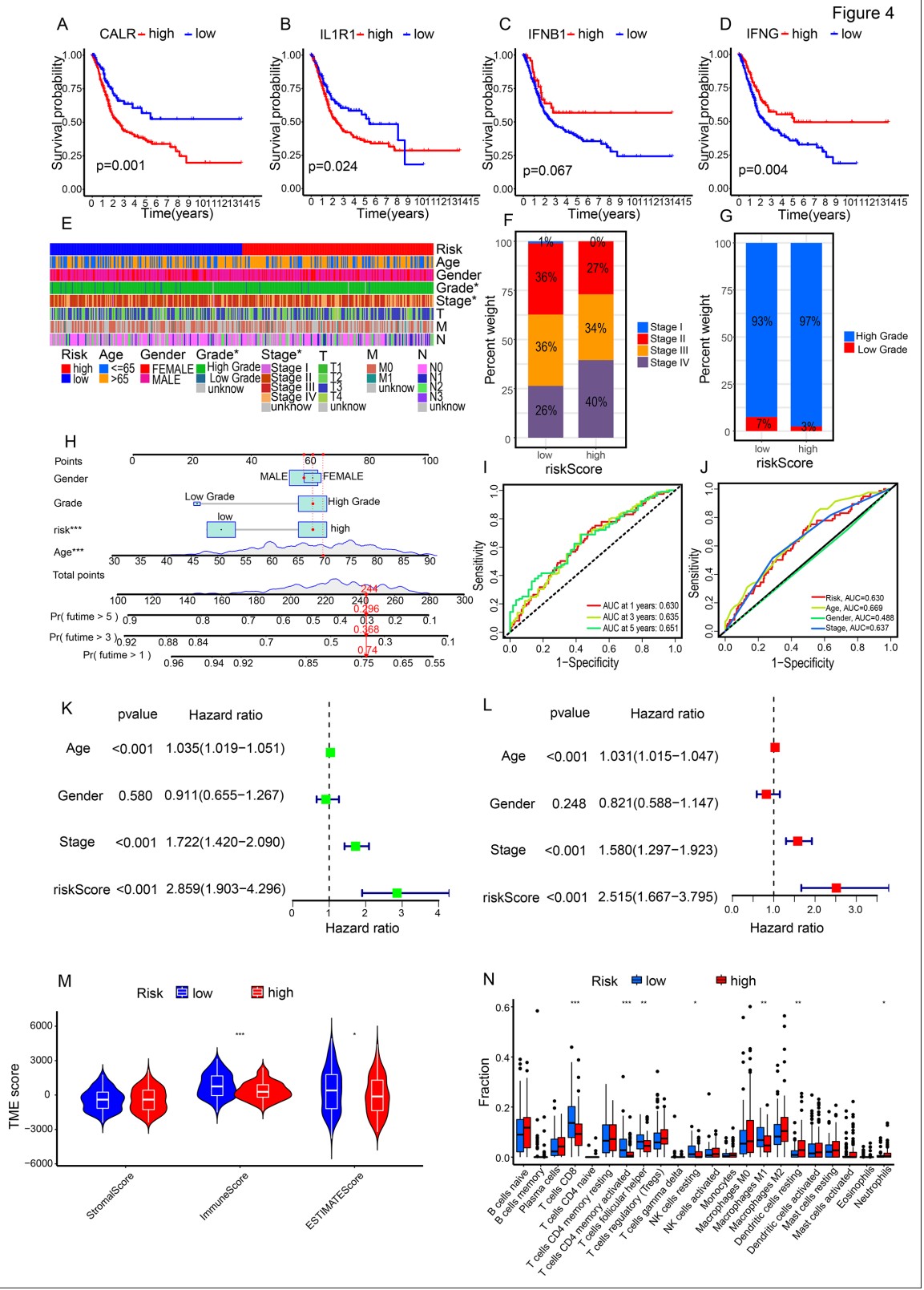

**Figure 4.** The association between risk scores and clinical features and immune characteristics. (**A–D**) The association between the four genes composing the risk scores and patient prognosis. (**E**) Clinical heatmap of the two subgroups. (**F, G**) Differences in stage (**F**) and grade (**G**) between the two subgroups. (**H–J**) Constructed nomogram (**H**) and ROC (Receiver Operating Characteristic Curve) curve (**I–J**). (**K, L**) Cox regression analysis. (**M**) Stromal scores, immune scores, and ESTIMATE scores of the high-risk and low-risk groups. (**N**) Immune infiltration in the two subgroups.

*Figure 4 continued on next page*

*Figure 4 continued*

The online version of this article includes the following figure supplement(s) for figure 4:

**Figure supplement 1.** Relationship between risk scores and different clinical features.

**Figure supplement 2.** The relationship between risk scores and immune-related pathways and immune infiltration.

**Figure supplement 3.** The association between risk scores and immune cells.

**Figure supplement 4.** The correlation of CALR, IFNG, and IL1R1 with immune cell infiltration.

**Figure supplement 5.** Pathway enrichment analysis and correlation with immune checkpoints of CALR, IFN1B, IFNG, IL1R1, and risk score.

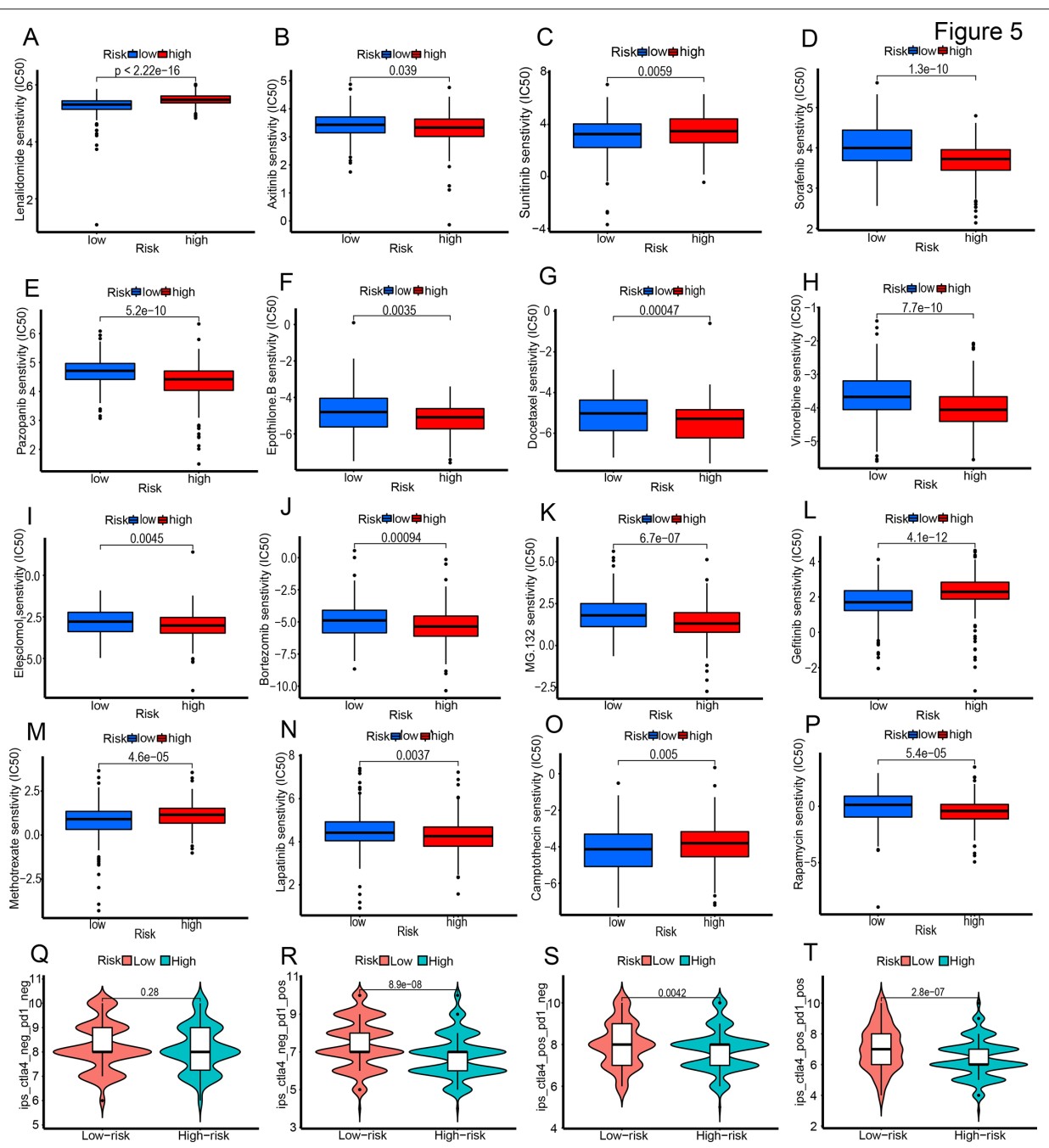

**Figure 5.** The association between the risk score and the sensitivity to chemotherapy (**A–P**) and immunotherapy (**Q–T**).

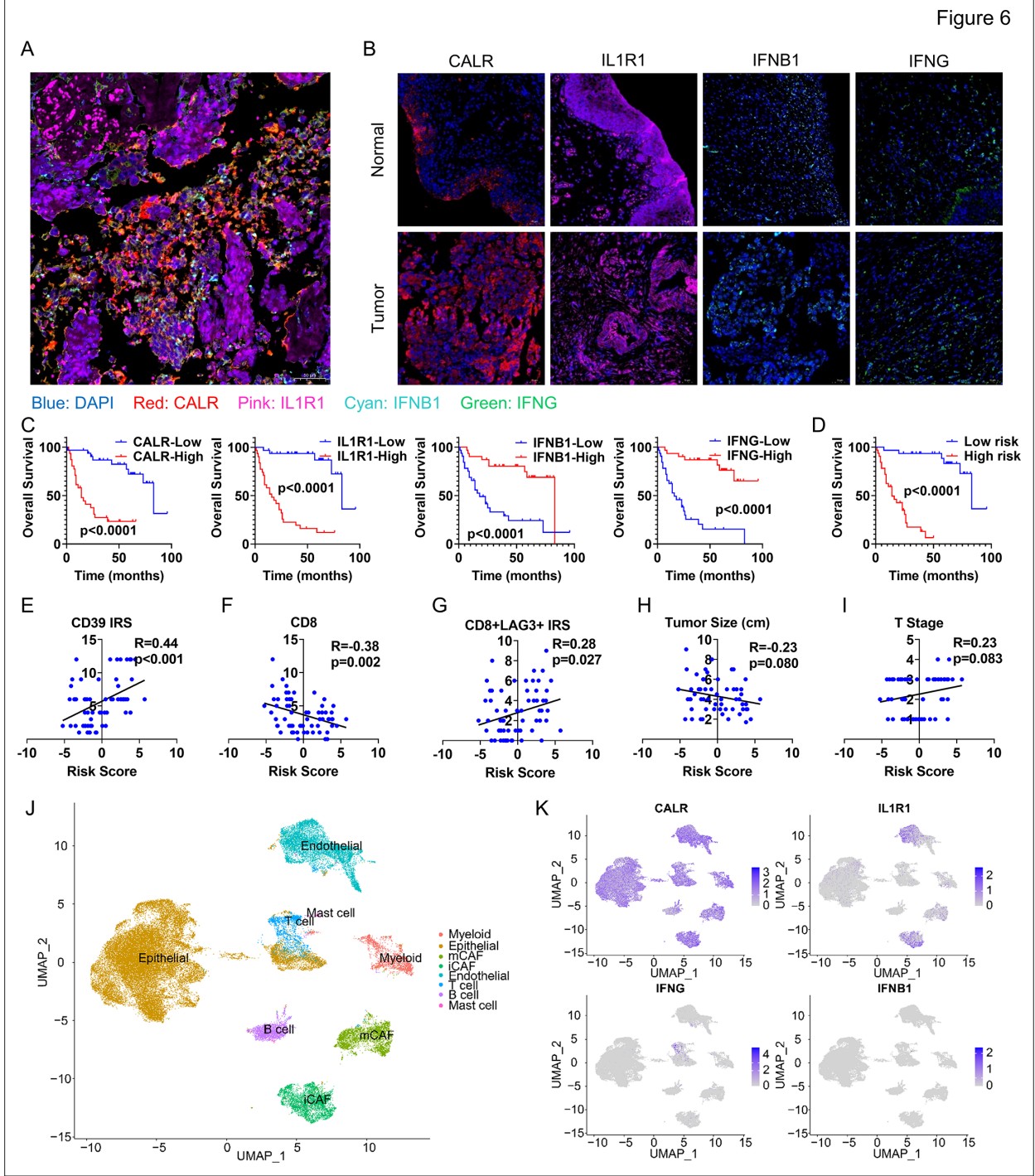

**Figure 6.** Validate the risk-scoring model through tissue arrays and single-cell sequencing. (**A, B**) Analyzing the expression of CALR, IFN1B, IFNG, and IL1R1 using immunofluorescence staining. The scale bar is 50 μm.(**C, D**) Survival analysis of the four genes (**C**) and risk scores (**D**) in patients. (**E–I**) The relationship between risk scores and CD39, CD8+, CD8+LAG3+, tumor size, and T stage. (**J, K**) Expression analysis of CALR, IFN1B, IFNG, and IL1R1 in single-cell sequencing data.

The online version of this article includes the following figure supplement(s) for figure 6:

**Figure supplement 1.** Expression of CALR, IFN1B, IFNG, and IL1R1 in tissues.

This suggests a crucial role of cancer-associated fibroblasts in remodeling the immune microenvironment of BLCA (*Figure 6J and K*, *Figure 6—figure supplement 1F and G*).

## Discussion

ICD, a regulated cell death type, can be triggered by internal or external antigens, activating an adaptive immune response (*Galluzzi et al., 2020*). Some chemotherapies, such as cisplatin and gemcitabine, can induce ICD, thereby activating the immune microenvironment to achieve antitumor effects (*Chen et al., 2023*; *McDonnell et al., 2015*). ICIs are a key focus in cancer research, blocking tumors by reactivating immune cells through targeting checkpoints like PD1/PD-L1 and CTLA4 (*Balar et al., 2021*). Combining chemotherapy with ICI treatment is expected to enhance immune function and inhibit tumors, given ICD's role in activating the immune microenvironment, and some studies targeting key genes in the ICD process or combining chemotherapy with immunotherapy have also achieved encouraging results (*Galluzzi et al., 2017*; *Jafari et al., 2020*; *Peng et al., 2021*). However, the results of clinical trials indicate that the response rate of BLCA to ICIs is significantly lower than expected. The tumor immunosuppressive microenvironment is undoubtedly an important influencing factor. Considering the complexity of the TME and the impact of ICD, we investigate and confirm the clinical utility of ICD-related genes using BLCA public databases, BLCA tissue microarrays, and single-cell sequencing data. This offers theoretical guidance for applying ICD in clinical settings.

Through unsupervised clustering analysis, we categorized TCGA-BLCA patients into two groups: ICD-high and ICD-low. Pathway enrichment analysis revealed that the ICD-high group predominantly clustered in pathways related to the immune system, consistent with previous literature reports (*Jiang et al., 2023*; *Kuang et al., 2023*). In the analysis of immune infiltration, we found that the ICD-high group showed higher immune cell infiltration, coupled with lower tumor purity, indicating a lower invasive ability of the tumor and an activated immune state (*Dong et al., 2023*). We checked if ICD boosts ICI effectiveness by studying the connection between ICD-related gene expression and immunotherapy response rates. Turns out, the ICD-high group showed a much better response compared to the ICD-low group. This suggests that ICI treatment could work better in people with high ICD. Thus, we created a risk-scoring model using four ICD-related genes (CALR, IL1R1, IFNB1, and IFNG) identified through Cox regression and LASSO regression analyses. This model helps classify BLCA and predict immune infiltration, prognosis, and response to immunotherapy. Of them, IFNB1 and IFNG, both belonging to the interferon family, play crucial roles in anti-tumor immunity (*Gaidt et al., 2021*; *Boehm et al., 1997*). IL1R1 is a receptor for IL1α and IL1β, a member of the interleukin family, and plays an immunosuppressive role in immune regulation (*Liu et al., 2023*). In ICD, the CALR calcium-binding protein moves to the cell membrane to aid antigen presentation, a widely accepted process (*Qian et al., 2022*). Yet, the general increase in CALR levels does not necessarily have a positive regulatory effect. Studies indicate that elevated CALR expression in breast cancer can enhance tumor cell metastasis and resistance to chemotherapy (*Liu et al., 2021*). Similarly, our study found that high CALR expression in BLCA has an inhibitory effect. Survival analysis showed that high CALR and IL1R1 expression was detrimental to patient survival, while high IFNB1 and IFNG expression was beneficial.

To verify our risk-scoring model, we used univariate and multivariate Cox regression analyses. The results indicate that the ICD scoring system can independently predict the prognosis of BLCA patients. Subgroup analysis based on clinical characteristics showed no difference between high-risk and low-risk groups in the early stages of the disease. However, in later stages, the high-risk group had shorter survival times. This may be due to changes in the TME at different disease stages, with a decrease in immune-killing molecules and an increase in inhibitory molecules as the tumor advances (*Li et al., 2021*). Additional analysis of the correlation between the risk scores and immune cell infiltration confirmed this: as the risk score increased, the infiltration of cells with anti-tumor effects decreased, while the infiltration of M2-type macrophages and cancer-associated fibroblasts increased. These findings confirm the efficacy of our risk-scoring model in assessing patients' immune infiltration, prognosis, and tumor progression.

In treating BLCA, chemotherapy, targeted therapy, and immunotherapy are effective but have limitations in sensitivity and specificity (*Tran et al., 2021*). Our risk signature predicts drug sensitivity. High-risk group responds better to chemotherapy, and low-risk group shows higher response in immunotherapy (PD1 or CTLA4). This matches the higher immune infiltration in the low-risk patient group, emphasizing that immune infiltration level is crucial for successful immunotherapy. However,

we find here that the quantity of immune cells in the tumor does not directly decide the patient's prognosis. This is because many infiltrating immune cells are functionally exhausted, and only immunotherapy can activate the immune response within the tumor, improving prognosis. This is confirmed in BLCA tissue microarray stains. Specifically, the risk score shows a negative correlation with CD8+T cell infiltration but a positive correlation with T cell exhaustion (CD8+LAG3+ or CD39). This shows our risk scoring model can categorize patients based on immune infiltration, specifically identifying a high-risk group (lower immune infiltration) and a low-risk group (higher immune infiltration). Personalized treatment is crucial for patients in this context. Additionally, in BLCA single-cell sequencing, we found IL1R1 is abundantly expressed in cancer-associated fibroblasts, linked to tumor promotion and immunosuppression (*Liao et al., 2019*). This suggests a crucial role for these fibroblasts in shaping the immune microenvironment of BLCA, guiding our next research.

## Conclusions

ICD, a type of cell death triggered in an immunogenic environment, influences immune response. Our study links ICD-related genes to BLCA prognosis and tumor immune infiltration. We created a validated risk-scoring model with four key genes (CALR, IL1R1, IFNB1, and IFNG) using TCGA-BLCA, GEO datasets, and tissue microarray staining. This model effectively assesses immune infiltration, prognosis, immunotherapy response, and drug sensitivity in BLCA, guiding personalized treatment and future research.

# Materials and methods

## Identification of differentially expressed ICD-related genes

ICDs were sourced from a previous article (*Garg et al., 2016*). We found differentially expressed genes (DEGs) in TCGA-BLCA using the 'Limma' R package. For TCGA samples: 421 tumor samples and 19 normal samples. Database release date: March 29, 2022, v36 versions. Coding package version: R version 4.1.1. This involved comparing gene expression in normal and tumor samples. DEGs interacting with ICDs were identified. Enrichment scores of ICDs were assessed using ssGSEA analysis with the 'GSVA' R package. Patients were then grouped as 'high' or 'low' based on these scores. Finally, DEGs were visualized through volcano and heatmaps using R packages 'pheatmap' and 'ggplot2'. All sample data are derived from the TCGA and The Cancer Immunome Atlas (TCIA) databases.

## Functional enrichment analysis

We compared ICD-high and ICD-low groups by analyzing GO and KEGG using the 'clusterProfiler' R package. GSEA assessed differences in expressed gene sets between the groups using the MSigDB collection.

## Analysis of somatic mutation

BLCA patient mutation data from TCGA were visualized using 'Maftools' in waterfall plots. Boxplots summarized tumor mutational burden (TMB), and the 20 most frequently mutated genes were shown in waterfall plots.

## Analysis of immune infiltration in the TME

Used the 'ESTIMATE' R package to predict the tumor microenvironment (TME) scores for BLCA samples and assess immunocyte proportions. XCELL, TIMER, QUANTISEQ, MCPcounter, EPIC, CIBERSORT, and CIBERSORT-ABS were used to explore immune cell infiltration, visualizing differences with a violin diagram and analyzing with the Wilcoxon signed-rank test. Evaluated immunotherapeutic response using the IPS algorithm and compared immune checkpoint and HLA-related gene expression in different groups. Also, accessed IPS data from The Cancer Immunome Atlas (TCIA) to compare responses to ICIs across BLCA patient groups.

## Construction and validation of risk-scoring model

To explore the prognostic value of ICD-related genes in BLCA. Using TCGA-BLCA data, we conducted Cox regression analysis and LASSO Cox regression to build a prognostic model. Risk score=CALR*(0.5378)+IFN1B*(−0.6349)+IFNG*(−0.2028)+IL1R1*(0.0919). Risk scores were calculated, and

patients were divided into low-risk and high-risk groups. We validated the model using the GSE13507 data and BLCA tissue arrays (HBlaU079Su01, Shanghai Outdo Biotech Co., Ltd) including a total of 63 cancer tissues and 16 cancer-adjacent normal tissues from patients with BLCA. Kaplan–Meier curves for overall survival (OS) in low-risk and high-risk groups were created with the R packages 'Survminer' and 'Survival'. Nomogram was created using the 'rms' package to integrate risk scores and other clinicopathological information to assess survival probability. ROC analysis was used to evaluate the nomogram's ability to predict survival. The R package 'pRRophetic' was used to evaluate the drug sensibility. The Remmele and Stegner's semiquantitative IRS scale was employed to assess the expression levels of each marker, as detailed in earlier study (*Remmele and Stegner, 1987*). The R umap package was utilized to conduct the UMAP analyses of the individual cells. A two-sided $p<0.05$ was considered valuable. The reagents used are as follows: anti-IFNB1 (cat#: bs-0787R, BIOSS), anti-IL1R1 (cat#: bs-20697R, BIOSS), anti-IFNG (cat#: MAB48116, Bioswamp), and anti-CALR (cat#: A1066, ABclonal). The general workflow for single-cell sequencing analysis includes sample preparation, RNA extraction, library construction, high-throughput sequencing, data preprocessing (quality control, normalization), clustering, cell type annotation, differential expression analysis, and functional enrichment.

## Acknowledgements

This research was supported by the National Natural Science Foundation of China (no. 82303623), the Noncommunicable Chronic Diseases-National Science and Technology Major Project (no. 2024ZD0532300), the Intramural Funding from Shanghai Public Health Clinical Center (no. KY-GW-2025-01), the Open Project of Key Laboratory of Vascular Aging (HUST) Ministry of Education (no. VAME-2024-1), the National Key Research and Development Project of China (no. 2019YFA0905600), the China Postdoctoral Science Foundation (no. 2023M731199), and the Postdoctoral Innovation Research Post of Hubei Provincial Department of Human Resources and Social Security (no. 331048). The authors acknowledge that the views expressed in this work are personal and may not necessarily represent the perspectives of official institutions.

## Additional information

### Funding

| Funder | Grant reference number | Author |
| --- | --- | --- |
| National Natural Science Foundation of China | 82303623 | Lilong Liu |
| Noncommunicable Chronic Diseases-National Science and Technology Major Project | 2024ZD0532300 | Yi Hao |
| Shanghai Public Health Clinical Center | Intramural funding KY-GW-2025-01 | Yi Hao |
| Open Project of Key Laboratory of Vascular Aging (HUST), Ministry of Education | VAME-2024-1 | Yi Hao |
| National Key Research and Development Program of China | 2019YFA0905600 | Yi Hao |
| China Postdoctoral Science Foundation | 2023M731199 | Lilong Liu |
| Hubei Provincial Department of Human Resources and Social Security | Postdoctoral Innovation Research Post 331048 | Lilong Liu |

| Funder | Grant reference number | Author |
|---|---|---|
| Noncommunicable Chronic Diseases-National Science and Technology Major Project | 2024ZD0525700 | Zheng Liu |

The funders had no role in study design, data collection and interpretation, or the decision to submit the work for publication.

## Author contributions
Lilong Liu, Data curation, Formal analysis, Funding acquisition, Writing – original draft, Writing – review and editing, Conceptualization, Project administration; Zhenghao Liu, Lei Fan, Zhipeng Yao, Data curation, Formal analysis, Writing – original draft, Writing – review and editing; Junyi Hu, Yaxin Hou, Yang Li, Yuhong Ding, Yingchun Kuang, Data curation, Writing – review and editing; Ke Chen, Zheng Liu, Conceptualization, Writing – review and editing; Yi Hao, Conceptualization, Funding acquisition, Writing – review and editing

## Author ORCIDs
Lilong Liu ⓘ https://orcid.org/0000-0001-5459-5204
Zhenghao Liu ⓘ https://orcid.org/0000-0003-1185-598X
Lei Fan ⓘ http://orcid.org/0009-0006-7338-0384
Zhipeng Yao ⓘ https://orcid.org/0009-0000-3104-2092
Junyi Hu ⓘ https://orcid.org/0000-0002-7062-6434
Yaxin Hou ⓘ https://orcid.org/0000-0002-1330-6506
Yang Li ⓘ https://orcid.org/0000-0003-4135-4334
Yuhong Ding ⓘ https://orcid.org/0000-0003-2071-656X
Yingchun Kuang ⓘ https://orcid.org/0009-0009-3596-1235
Ke Chen ⓘ https://orcid.org/0000-0002-2098-1921
Yi Hao ⓘ https://orcid.org/0000-0002-0339-9905
Zheng Liu ⓘ https://orcid.org/0000-0001-9193-9467

Reviewer #1 (Public review): https://doi.org/10.7554/eLife.95326.3.sa1
Author response https://doi.org/10.7554/eLife.95326.3.sa2

---

# Additional files

## Supplementary files
MDAR checklist

Source code 1. Code used for the analysis in this paper.

## Data availability
This study did not generate new experimental data. All primary data were obtained from TCGA https://portal.gdc.cancer.gov/ and GEO (https://www.ncbi.nlm.nih.gov/geo/query/acc.cgi?acc=GSE13507), and the single-cell sequencing data https://ngdc.cncb.ac.cn/gsa-human/browse/HRA000212 or https://www.ncbi.nlm.nih.gov/bioproject/PRJNA662018/, with all generated and analyzed data fully included in the manuscript and supporting files.

The following previously published datasets were used:

| Author(s) | Year | Dataset title | Dataset URL | Database and Identifier |
|---|---|---|---|---|
| Kim WJ, Kim EJ, Kim SK, Kim YJ | 2010 | Predictive value of progression-related gene classifier in primary non-muscle invasive bladder cancer | https://www.ncbi.nlm.nih.gov/geo/query/acc.cgi?acc=GSE13507 | NCBI Gene Expression Omnibus, GSE13507 |

*Continued on next page*

*Continued*

| Author(s) | Year | Dataset title | Dataset URL | Database and Identifier |
|---|---|---|---|---|
| Chen Z, Zhou L, Liu L, Hou Y, Xiong M, Yang Y, Hu J, Chen K | 2020 | Single Cell RNA Sequencing Highlights the Role of Inflammatory Cancer-associated Fibroblasts in Bladder Urothelial Carcinoma | https://www.ncbi.nlm.nih.gov/bioproject/PRJNA662018/ | NCBI BioProject, PRJNA662018 |
| Chen K | 2020 | Single cell RNA sequencing of bladder urothelial carcinoma | https://ngdc.cncb.ac.cn/gsa-human/browse/HRA000212 | Genome Sequence Archive, PRJCA002909 |

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
