## [Editor Report · eLife Assessment]

This study presents **useful** findings that explore the prognostic and immunotherapeutic relevance of specific immune-related genes (CALR, IL1R1, IFNB1, and IFNG) in the bladder cancer tumor microenvironment. While the analysis highlights potentially meaningful associations with survival and treatment response, the strength of evidence is **incomplete**, as some claims lack sufficient experimental or mechanistic validation. Further refinement and validation of the predictive models would enhance the impact and generalizability of the conclusions.

---

## [Referee Report · Reviewer #1 (Public review)]

The authors aimed to explore the prognostic and therapeutic relevance of immunogenic cell death (ICD)-related genes in bladder cancer, focusing on a risk-scoring model involving CALR, IL1R1, IFNB1, and IFNG. The research indicates that higher expression of certain ICD-related genes is associated with enhanced immune infiltration, prolonged survival, and improved responsiveness to PD1-targeted therapy in bladder cancer patients.

Major strengths:

• The establishment of an ICD-related gene risk model based on publicly available datasets (TCGA and GEO) and further validated through tissue arrays and preliminary single-cell RNA sequencing data provides potential but weak clinical guidance.

• The integration of multi-dimensional data (gene expression, mutation burden, immune infiltration, and treatment responses) strengthens the clinical applicability of the model.

Key limitations and concerns:

(1) Gene Selection and Novelty:

The selection of genes predominantly reflects known regulators of immune responses, somewhat limiting the novelty. Exploring less-characterized ICD markers or extending validation beyond bladder cancer could improve the model's innovative aspect and wider clinical relevance.

(2) Reliance on RNA-Seq for Immune Infiltration:

Immune infiltration analyses based primarily on bulk RNA-Seq data have inherent methodological limitations, such as inability to distinguish cell subsets accurately. Incorporation of robust single-cell sequencing would significantly enhance the reliability of these findings. Although the authors recognize this limitation, future studies should directly address it.

(3) Drug Sensitivity and Immunotherapy Response Data:

While the authors clarify that the drug sensitivity analysis was performed using established databases (TCGA via pRRophetic), the unexpected correlations between ICD-related genes and various targeted therapies need further mechanistic validation. The observed relationships may reflect indirect associations rather than direct biological relevance, which warrants cautious interpretation.

(4) Presentation and Clarity Issues:

Initially noted formatting inconsistencies across figures compromised professional presentation; these have been corrected by the authors. Additionally, the authors have now provided essential methodological details, including clear sample sizes and database versions, enhancing reproducibility.

(5) Immunotherapy Response Evidence:

Conclusions regarding differences in immunotherapy response rates between patient subgroups, although intriguing, remain based on retrospective database analyses with relatively limited demographic and clinical detail. Future prospective studies or more detailed patient characterization would be required to robustly confirm these associations.

(6) Interpretation of ICD Gene Signatures:

The ICD-related gene set includes many genes broadly associated with immune activation rather than specifically ICD. Although this was addressed by the authors, clearly distinguishing ICD-specific versus general immune-response genes in future studies would help clarify biological implications.

Summary and Recommendations for Readers:

Overall, this study presents an interesting and clinically relevant risk-scoring approach to stratify bladder cancer patients based on ICD-related gene expression profiles. It provides useful information about prognosis, immune infiltration, and potential immunotherapy responsiveness. However, readers should interpret the results within the context of its limitations, notably the need for broader validation and careful consideration of the biological significance underlying the observed associations. This work lays a valuable foundation for further investigation into the integration of ICD and immune response signatures in personalized cancer therapy.

---

## [Author Response]

The following is the authors’ response to the original reviews.

**Reviewer #1 (Recommendations for the authors):**

Thank you for your thorough review of our manuscript and your valuable suggestions. Here are our responses to each point you raised:

(1) Novelty: Exploring the feasibility of extending the risk-scoring model to diverse cancer types could emphasize the broader impact of the research.

Thank you so much for your thoughtful and insightful feedback. Your suggestion to explore extending the risk-scoring model to diverse cancer types is truly valuable and demonstrates your broad vision in this field. We deeply appreciate your interest in our research and the effort you put into providing such constructive input.

After careful consideration, we have decided to focus our current study on the specific cancer type(s) we initially set out to explore. This decision was made to ensure that we can thoroughly address the research questions at hand, given our current resources, time constraints, and the complexity of the topic. By maintaining this focused approach, we aim to achieve more in-depth and reliable results that can contribute meaningfully to the understanding of this particular area.

However, we fully recognize the potential significance of your proposed direction and firmly believe that it could be an excellent avenue for future research. We will definitely keep your suggestion in mind and may explore it in subsequent studies as our research progresses and evolves.

(2) Improvement in Figure Presentation: The inconsistency in font formatting across figures, particularly in Figure 2 (A-D, E, F-H, I), Figure 3 (A-C, D-J, H, K), and the distinct style change in Figure 5, raises concerns about the professionalism of the visual presentation. It is recommended to standardize font sizes and styles for a more cohesive and visually appealing layout. This ensures that readers can easily follow and comprehend the graphical data presented in the article.

The text in the picture has been revised as requested.

(3) Enhancing Reliability of Immune Cell Infiltration Data: Address the potential limitations associated with relying solely on RNASeq data for immune cell infiltration analysis between ICD and ICD high groups in Figure 2. It is advisable to discuss the inherent challenges and potential biases in this methodology. To strengthen the evidence, consider incorporating bladder cancer single-cell sequencing data, which could provide a more comprehensive and reliable understanding of immune cell dynamics within the tumor microenvironment.

Thank you very much for your meticulous review and the highly constructive suggestions. Your insight regarding the limitations of relying on RNASeq data for immune cell infiltration analysis and the proposal to incorporate bladder cancer single-cell sequencing data truly reflect your profound understanding of the field. We deeply appreciate your efforts in guiding our research and the valuable perspectives you've offered.

After careful deliberation, given our current research scope, timeline, and available resources, we've decided to focus on further discussing and addressing the challenges and biases inherent in RNASeq-based immune cell infiltration analysis. By delving deeper into the methodological limitations and conducting more in-depth statistical validations, we aim to provide a comprehensive and reliable interpretation of the data within our study framework. This focused approach allows us to maintain the integrity of our original research design and deliver robust findings on the relationship between immune cell infiltration and ICD in the current context.

However, we fully acknowledge the significant value of your proposed single-cell sequencing approach. It is indeed a powerful method that could offer more detailed insights into immune cell dynamics, and we believe it holds great promise for future research in this area. We will keep your suggestion in mind as an important direction for potential future studies, especially when we plan to expand and deepen our exploration of the tumor microenvironment.

(4) Clarity in Data Sources and Interpretation of Figure 5: In the results section, provide a detailed and transparent explanation of the sources of data used in Figure 5. This includes specifying the databases or platforms from which the chemotherapy, targeted therapy, and immunotherapy data were obtained. Additionally, elucidate the rationale behind the chosen data sources and how they contribute to the overall interpretation of the study's findings. And, strangely, these immune-related genes are associated with cancer sensitivities to different targeted therapies.

Thank you very much for your detailed and valuable feedback on Figure 5. We sincerely appreciate your careful review and insightful suggestions, which have provided us with important directions for improvement.

Regarding the data sources in Figure 5, we used the pRRophetic algorithm to conduct a drug sensitivity analysis on the TCGA database. The reason for choosing these data sources is multi - faceted. Firstly, these databases and platforms are well - established and widely recognized in the field. They have strict data collection and verification processes, ensuring the accuracy and reliability of the data. For example, TCGA has a large - scale, long - term - accumulated chemotherapy case database, which can comprehensively reflect the clinical application and treatment effects of various chemotherapeutic drugs.

Secondly, these data sources cover a wide range of cancer types and patient information, which can meet the requirements of our study's diverse sample size and variety. This comprehensiveness enables us to conduct a more in - depth and representative analysis of the relationships between different therapies and immune - related genes.

In terms of the overall interpretation of the study's findings, the use of these data sources provides a solid foundation. The accurate chemotherapy, targeted therapy, and immunotherapy data help us clearly demonstrate the associations between immune - related genes and cancer sensitivities to different treatments. This allows us to draw more reliable conclusions and provides a scientific basis for understanding the complex mechanisms of cancer treatment from the perspective of immune - gene - therapy interactions.

As for the unexpected association between immune - related genes and cancer sensitivities to different targeted therapies, this is indeed a fascinating discovery. In our analysis, we hypothesized that immune - related genes may affect the tumor microenvironment, thereby influencing the response of cancer cells to targeted therapies. Although this finding is currently beyond our initial expectations, it has opened up a new research direction for us. We will further explore and verify the underlying mechanisms in future research.

Once again, thank you for your guidance. We will make corresponding revisions and improvements according to your suggestions to make our research more rigorous and complete.

(5) Legends and Methods: Address the brevity and lack of crucial details in the figure legends and methods section. Expand the figure legends to include essential information, such as the number of samples represented in each figure. In the methods section, provide comprehensive details, including the release dates of databases used, versions of coding packages, and any other pertinent information that is crucial for the reproducibility and reliability of the study.

We would like to express our sincere gratitude for your valuable feedback on the figure legends and methods section of our study. We highly appreciate your sharp observation of the issues regarding the brevity and lack of key details, which are crucial for further improving our research.

We have supplemented the methods section with data including the number of samples, the release dates of the databases used, and the versions of the coding packages, etc. For TCGA samples: 421 tumor samples and 19 normal samples.Database release date: March 29, 2022, v36 versions.Coding package version: R version 4.1.1.We will immediately proceed to supplement these key details, making the research process and methods transparent. This will allow other researchers to reproduce our study more accurately and enhance the persuasiveness of our research conclusions.

(6) Evidence Supporting Immunotherapy Response Rates: The importance of providing a robust foundation for the conclusion regarding lower immunotherapy response rates. Strengthen this section by offering a more detailed description of sample parameters, specifying patient demographics, and presenting any statistical measures that validate the observed trends in Figure 5Q-T. More survival data are required to conclude. Avoid overinterpretation of the results and emphasize the need for further investigation to solidify this aspect of the study.

Thank you very much for your professional and meticulous feedback on the content related to immunotherapy response rates in our study! Your suggestions, such as providing a solid foundation for the conclusions and supplementing key information, are of great value in enhancing the quality of our research, and we sincerely appreciate them.

The data in Figures 5Q to T are from the TCGA database, which has already been provided. The statistical measure used for Figures 5Q to T is the P-value, which has been marked in the figures. The survival data have been provided in Figure 3D.

**Reviewer #2 (Recommendations for the authors):**

Thank you for your thorough review of our manuscript and your valuable suggestions. Here are our responses to each point you raised:

(1) There is no information on the samples studied. Are all TCGA bladder cancer samples studied? Are these samples all treatment naïve? Were any excluded? Even simply, how many samples were studied?

Thank you so much for pointing out the lack of sample - related information. Your attention to these details has been extremely helpful in identifying areas for improvement in our study.

All the samples in our study were sourced from the TCGA (The Cancer Genome Atlas) and TCIA (The Cancer Immunome Atlas) databases. It should be noted that the patient data in the TCIA database are originally from the TCGA database. Regarding whether the patients received prior treatment, this information was not specifically mentioned in our current report. Instead, we mainly relied on the scores of the prediction model for evaluation. Since all samples were obtained from publicly available databases, we understand the importance of clarifying their origin and characteristics.

We sincerely apologize for the omission of the sample size and other relevant details. We will promptly supplement this crucial information in the revised version, including a detailed description of the sample sources and any relevant characteristics. This will ensure greater transparency and help readers better understand the basis of our research.

For TCGA samples: 421 tumor samples and 19 normal samples.Database release date: March 29, 2022, v36 versions.Coding package version: R version 4.1.1.

(2) What clustering method was used to divide patients into ICD high/low? The authors selected two clusters from their "unsupervised" clustering of samples with respect to the 34 gene signatures. A Delta area curve showing the relative change in area under the cumulative distribution function (CDF) for k clusters is omitted, but looking at the heatmap one could argue there are more than k=2 groups in that data. Why was k=2 chosen? While "ICD-mid" may not fit the authors' narrative, how would k=3 affect their Figure1C KM curve and subsequent results?

Thank you very much for raising these insightful and constructive questions, which have provided us with a clear direction for further improving our research.

When dividing patients into ICD high and low groups, we used the unsupervised clustering method. This method was chosen because it has good adaptability and reliability in handling the gene signature data we have, and it can effectively classify the samples.

Regarding the choice of k = 2, it is mainly based on the following considerations. Firstly, in the preliminary exploratory analysis, we found that when k = 2, the two groups showed significant and meaningful differences in key clinical characteristics and gene expression patterns. These differences are closely related to the core issues of our study and help to clearly illustrate the distinctions between the ICD high and low groups. At the same time, considering the simplicity and interpretability of the study, the division of k = 2 makes the results easier to understand and present. Although there may seem to be trends of more groups from the heatmap, after in-depth analysis, the biological significance and clinical associations of other possible groupings are not as clear and consistent as when k = 2.

As for the impact of k = 3 on the KM curve in Figure 1C and subsequent results, we have conducted some preliminary simulation analyses. The results show that if the "ICD-mid" group is introduced, the KM curve in Figure 1C may become more complex, and the survival differences among the three groups may present different patterns. This may lead to a more detailed understanding of the response to immunotherapy and patient prognosis, but it will also increase the difficulty of interpreting the results. Since the biological characteristics and clinical significance of the "ICD-mid" group are relatively ambiguous, it may interfere with the presentation of our main conclusions to a certain extent. Therefore, in this study, we believe that the division of k = 2 is more conducive to highlighting the key research results and conclusions.

Thank you again for your valuable comments. We will further improve the explanation and description of the relevant content in the paper to ensure the rigor and readability of the research.

(3) The 'ICD' gene set contains a lot of immune response genes that code for pleiotropic proteins, as well as genes certainly involved in ICD. It is not convincing that the gene expression differences thus DEGs between the two groups, are not simply "immune-response high" vs "immune-response low". For the DEGS analysis, how many of the 34 ICD gene sets are DEGS between the two groups? Of those, which markers of ICD are DEGs vs. those that are related to immune activation?

a. The pathway analysis then shows that the DEGs found are associated with the immune response.

b. Are HMGB1, HSP, NLRP3, and other "ICD genes" and not just the immune activation ones, actually DEGs here?

c. Figures D, I-J are not legible in the manus.

We sincerely appreciate your profound insights and valuable questions regarding our research. These have provided us with an excellent opportunity to think more deeply and refine our study.

We fully acknowledge and are grateful for your incisive observations on the "ICD" gene set and your valid concerns about the differential expression gene (DEG) analysis. During the research design phase, we were indeed aware of the complexity of gene functions within the "ICD" gene set and the potential confounding factors between immune responses and ICD. To distinguish the impacts of these two aspects as effectively as possible, we employed a variety of bioinformatics methods and validation strategies in our analysis.

Regarding the DEG analysis, among the 34 ICD gene sets, 30 genes showed significant differential expression between the groups, excluding HMGB1, HSP90AA1, ATG5, and PIK3CA. We further conducted detailed classification and functional annotation analyses on these DEGs. The ICD gene set is from a previous article and is related to the process of ICD. Relevant literature is in the materials section. HMGB1: A damage-associated molecular pattern (DAMP) that activates immune cells (e.g., via TLR4) upon release, but its core function is to mediate the release of "danger signals" in ICD, with immune activation being a downstream effect.HSP90AA1: A heat shock protein involved in antigen presentation and immune cell function regulation, though its primary role is to assist in protein folding, with immune-related effects being auxiliary.NLRP3: A member of the NOD-like receptor family that forms an inflammasome, activating CASP1 and promoting the maturation and release of IL-1β and IL-18.Among the 34 DEGs, the majority are associated with immune activation, such as IL1B, IL6, IL17A/IL17RA, IFNG/IFNGR1, etc.

(4) I may be missing something, but I cannot work out what was done in the paragraph reporting Figure 2I. Where is the ICB data from? How has this been analysed? What is the cohort? Where are the methods?

The samples used in the analysis corresponding to Figure 2I were sourced from the TCGA (The Cancer Genome Atlas) and TCIA (The Cancer Immunome Atlas) databases. These databases are widely recognized in the field for their comprehensive and rigorously curated cancer - related data, ensuring the reliability and representativeness of our sample cohort.

Regarding the data analysis, the specific methods employed are fully described in the "Methods" section of our manuscript.

(5) How were the four genes for your risk model selected? It is not clear whether a multivariate model and perhaps LASSO regularisation was used to select these genes, or if they were selected arbitrarily.

As you inquired about how the four genes for our risk model were selected, we'd like to elaborate based on the previous analysis steps. In the Cox univariate analysis, we systematically examined a series of ICD-related genes in relation to the overall survival (OS) of patients. Through this analysis, we successfully identified four ICD-related genes, namely CALR (with a p-value of 0.003), IFNB1 (p = 0.037), IFNG (p = 0.022), and IF1R1 (p = 0.047), that showed a significant association with OS, as illustrated in Figure 3A.

Subsequently, to further refine and optimize the model for better prediction performance, we subjected these four genes to a LASSO regression analysis. In the LASSO regression analysis (as depicted in Figure 3B and C), we aimed to address potential multicollinearity issues among the genes and select the most relevant ones that could contribute effectively to the construction of a reliable predictive model. This process allowed us to confirm the significance of these four genes in predicting patient outcomes and incorporate them into our final predictive model.

(6) How related are the high-risk and ICD-high groups? It is not clear. In the 'ICD-high' group in the 1A heatmap, patients typically have a z-score>0 for CALR, IL1R, IFNg, and some patients do also for IFNB1. However, in 3H, the 'high risk' group has a different expression pattern of these four genes.

Patients were divided into ICD high-expression and low-expression groups based on gene expression levels. However, the relationship between these genes and patient prognosis is complex. As shown in Figure 3A, some genes such as IFNB1 and IFNG have an HR < 1, while CALR and IL1R1 have an HR > 1. Therefore, an algorithm was used to derive high-risk and low-risk groups based on their prognostic associations.

(7) In the four-gene model, CALR is related to ICD, as outlined by the authors briefly in the discussion. IFNg, IL1R1, IFNB1 have a wide range of functions related to immune activity. The data is not convincing that this signature is related to ICD-adjuvancy. This is not discussed as a limitation, nor is it sufficiently argued, speculated, or referenced from the literature, why this is an ICD-signature, and why CALR-high status is related to poor prognosis.

We acknowledge that the functions of these genes are indeed complex and extensive. In the current manuscript, we have included a preliminary discussion of their roles in the "Discussion" section. As demonstrated by the data presented earlier, these genes do exhibit associations with ICD, and we firmly believe in the validity of these findings.

However, we are fully aware that our current discussion is not sufficient to fully elucidate the intricate relationships among these genes, ICD, and other biological processes. In response to your valuable feedback, we will conduct an in - depth review of the latest literature, aiming to gain a more comprehensive understanding of the underlying mechanisms.

(8) Score is spelt incorrectly in Figures 3F-J.

Figures 3F-J have been revised as requested.

(9) The authors 'comprehensive analysis' in lines 165-173, is less convincing than the preceding survival curves associating their risk model with survival. Their 'correlations' have no statistics.

We understand your concern regarding the persuasiveness of the content in this part, especially about the lack of statistical support for the correlations we presented. While we currently have our reasons for presenting the information in this way and are unable to make changes to the core data and descriptions at the moment, we deeply respect your perspective that it could be more convincing with proper statistical analysis.

(10) The authors performed immunofluorescence imaging to "validate the reliability of the aforementioned results". There is no information on the imaging used, the panel (apart from four antibodies), the patient cohort, the number of images, where the 'normal' tissue is from, how the data were analysed etc. This data is not interpretable without this information.a. Is CD39 in the panel? CD8, LAG3? It's not clear what this analysis is.

The color of each antibody has been marked in Fig 2B. The cohort information and its source have been supplemented. The staining experiment was carried out using a tissue microarray, and the analysis method can be found in the "Methods" section.Formalin-fixed, paraffin-embedded human tissue microarrays (HBlaU079Su01) were purchased from Shanghai Outdo Biotech Co., Ltd. (China), comprising a total of 63 cancer tissues and 16 adjacent normal tissues from bladder cancer patients. Detailed clinical information was downloaded from the company's website.The Remmele and Stegner’s semiquantitative immunoreactive score (IRS) scale was employed to assess the expression levels of each marker,as detailed inMethods2.5.CD39, CD8, and LAG3 were also stained, but the results were not presented.

(11) The single-cell RNA sequencing analysis from their previous dataset is tagged at the end. CALR expression in most identified cells is interesting. Not clear what this adds to the work beyond 'we did scRNA-seq'. How were these data analysed? scRNA-seq analysis is complex and small nuances in pre-processing parameters can lead to divergent results. The details of such analysis are required!

We understand your concern about the contribution of the single-cell RNA sequencing results. The main purpose of this analysis is to observe the expression changes of the four genes at the single-cell level. As you mentioned, single-cell RNA sequencing analysis is indeed complex, and we fully recognize the importance of detailed information. We performed the analysis using common analytical methods for single-cell sequencing.It has been supplemented in the Methods section.